# Human Placental Mesenchymal Stem Cells and Derived Extracellular Vesicles Ameliorate Lung Injury in Acute Respiratory Distress Syndrome Murine Model

**DOI:** 10.3390/cells12232729

**Published:** 2023-11-29

**Authors:** Paulius Valiukevičius, Justinas Mačiulaitis, Dalia Pangonytė, Vitalija Siratavičiūtė, Katarzyna Kluszczyńska, Ugnė Kuzaitytė, Rūta Insodaitė, Ieva Čiapienė, Ramunė Grigalevičiūtė, Vilma Zigmantaitė, Astra Vitkauskienė, Romaldas Mačiulaitis

**Affiliations:** 1Faculty of Medicine, Medical Academy, Lithuanian University of Health Sciences, 44307 Kaunas, Lithuania; ugne.kuzaityte@lsmu.lt; 2Institute of Physiology and Pharmacology, Faculty of Medicine, Medical Academy, Lithuanian University of Health Sciences, 44307 Kaunas, Lithuania; justinas.maciulaitis@lsmu.lt (J.M.); ruta.insodaite@lsmu.lt (R.I.); romaldas.maciulaitis@lsmu.lt (R.M.); 3Laboratory of Cardiac Pathology, Institute of Cardiology, Lithuanian University of Health Sciences, 44307 Kaunas, Lithuania; dalia.pangonyte@lsmu.lt (D.P.); vitalija.sirataviciute@lsmu.lt (V.S.); 4Department of Molecular Biology of Cancer, Medical University of Lodz, 90-419 Lodz, Poland; katarzyna.kluszczynska@umed.lodz.pl; 5Institute of Cardiology, Medical Academy, Lithuanian University of Health Sciences, 44307 Kaunas, Lithuania; ieva.ciapiene@lsmu.lt; 6Biological Research Center, Lithuanian University of Health Sciences, 44307 Kaunas, Lithuania; ramune.grigaleviciute@lsmu.lt (R.G.); vilma.zigmantaite@lsmu.lt (V.Z.); 7Department of Laboratory Medicine, Faculty of Medicine, Medical Academy, Lithuanian University of Health Sciences, 44307 Kaunas, Lithuania; astra.vitkauskiene@lsmuni.lt

**Keywords:** acute respiratory distress syndrome (ARDS), mesenchymal stem cells (MSCs), extracellular vesicles (EVs), exosomes, preclinical model, inflammation, immunomodulation, COVID-19, good manufacturing practice (GMP)

## Abstract

This study investigates the therapeutic potential of human placental mesenchymal stem cells (P-MSCs) and their extracellular vesicles (EVs) in a murine model of acute respiratory distress syndrome (ARDS), a condition with growing relevance due to its association with severe COVID-19. We induced ARDS-like lung injury in mice using intranasal LPS instillation and evaluated histological changes, neutrophil accumulation via immunohistochemistry, bronchoalveolar lavage fluid cell count, total protein, and cytokine concentration, as well as lung gene expression changes at three time points: 24, 72, and 168 h. We found that both P-MSCs and EV treatments reduced the histological evidence of lung injury, decreased neutrophil infiltration, and improved alveolar barrier integrity. Analyses of cytokines and gene expression revealed that both treatments accelerated inflammation resolution in lung tissue. Biodistribution studies indicated negligible cell engraftment, suggesting that intraperitoneal P-MSC therapy functions mostly through soluble factors. Overall, both P-MSC and EV therapy ameliorated LPS-induced lung injury. Notably, at the tested dose, EV therapy was more effective than P-MSCs in reducing most aspects of lung injury.

## 1. Introduction

Acute respiratory distress syndrome (ARDS) is a serious and potentially fatal condition characterized by lung inflammation, impaired gas exchange, and acute respiratory failure [1]. It is a clinical diagnosis based on the Berlin criteria for adults [2]. ARDS is most commonly associated with pneumonia or non-pulmonary sepsis; however, other causes may lead to ARDS as well, such as the aspiration of gastric contents, pancreatitis, and chemotherapy [3]. The current approved ARDS treatment options are mostly supportive, consisting of lung-protective ventilation, prone positioning, and broad-acting systemic corticosteroids or neuromuscular blockers. Other clinically tested pharmacological interventions have not been approved for use [4]. During the COVID-19 pandemic, ARDS was recognized as one of the most serious complications for patients infected with SARS-CoV-2, since up to a third of patients hospitalized with COVID-19 developed ARDS [5], which resulted in an average 90-day mortality of 31% [6]. The causal agent of COVID-19, SARS-CoV-2, has been identified for its proclivity to trigger hyperinflammatory reactions within lung tissue [7]. The overwhelming incidence of ARDS as a severe manifestation of COVID-19 in the aftermath of the worldwide pandemic has highlighted the importance of discovering effective therapies for severe respiratory complications.

ARDS is the result of a dysregulated pulmonary inflammatory response to a variety of aforementioned triggers. The initial insult triggers the release of pro-inflammatory cytokines, such as interleukin-1β (IL-1β), tumor necrosis factor-alpha (TNF-α), and interleukin-8 (IL-8) [8]. These cytokines activate lung endothelial cells, which causes an increase in alveolar permeability. Furthermore, circulating leukocytes are recruited, and neutrophils, macrophages, and lymphocytes migrate into the alveolar space. These cells, predominantly neutrophils [8], release additional pro-inflammatory mediators and directly contribute to tissue injury via the release of reactive-oxygen-species-generating enzymes such as myeloperoxidase (MPO) [9,10]. Most research on ARDS pathophysiology has been carried out in various animal models. A common approach is the induction of sterile inflammation using endotoxins, such as lipopolysaccharides (LPSs). Usually, LPS is instilled intratracheally into the lungs; however, Khadangi et al. have found that intranasal LPS administration is just as effective [11]. In 2011, the American Thoracic Society described four main features of lung injury, of which at least three should be present in an ARDS animal model: histological evidence of lung injury, alteration of the alveolar–capillary barrier, an inflammatory response, and evidence of physiological dysfunction [12].

Mesenchymal stem cell (MSC) therapy is now recognized as a promising treatment option for ARDS due to its immunomodulatory, anti-inflammatory, and tissue repair properties, which could help restore the balance necessary for recovery. MSCs elicit their effects via both cell-to-cell interactions and the secretion of bioactive factors [13]. The MSC secretome consists of both soluble factors and extracellular vesicles (EVs). The soluble factors can be grouped into RNAs, cytokines, chemokines, and growth factors [14]. Experiments with conditioned media from MSC cultures have shown that most MSC therapeutic effects are derived from their secretome [15]. In fact, the secretome alone showed therapeutic potency in a murine lung injury model that closely resembled that of MSC administration [16]. A large part of the secretome is made up of exosomes, which are a subcategory of extracellular vesicles that are 50–150 nm in size and can carry a diverse cargo of nucleic acids, lipids, proteins, etc. [17,18]. Substantial research has revealed the promise of MSC-derived exosomes as cell-free therapies, with advantages such as lower immunogenicity, increased stability and safety, and the capacity to traverse biological barriers [18,19]. While encouraging results have been reported in many preclinical models of lung damage [20,21], as well as in some clinical trials [22,23], the specific mechanisms underlying the therapeutic effects of MSC-derived exosomes in ARDS-like conditions demand more extensive exploration.

The significance of establishing standardized and controlled manufacturing processes for MSCs cannot be overstated. After isolation, MSCs require significant in vitro expansion to meet clinical dosage requirements [24]. Elements like isolation procedure, cell density during plating, doubling times, passage numbers, and confluency profoundly influence MSC growth and function [24,25]. This emphasizes the critical need for uniformity in MSC production methods. Autologous MSCs present logistical challenges, especially for acute diseases like acute respiratory distress syndrome, due to manufacturing and quality release delays [26]. Allogeneic MSCs, often used as ‘off-the-shelf’ products, are generally hypo-immunogenic but can trigger donor-specific immune responses [27]. Factors like age, health status, and the disease severity of donors affect MSC properties, including potential karyotypic abnormalities. The extensive characterization of the manufacturing process minimizes the unexpected yield of cell quality from the donor material [27,28]. Therefore, carefully controlled manufacturing processes are essential for the production of safe and effective MSCs and should be established as early as possible to ensure better translation in clinical trials [29].

In the realm of MSC and their extracellular vesicle (EV) therapy for ARDS, numerous critical questions persist, necessitating further investigation. These include the absence of standardized production methods, a lack of comparative studies, limited insights into biodistribution patterns, ambiguity in defining appropriate study time points, a deficiency in quality control procedures, and a need for more comprehensive research on exosome isolation techniques and storage.

In this study, we induced ARDS in mice and evaluated the potential efficacy of placental mesenchymal stem cells (P-MSCs) and their EV in treating induced lung injury. We then analyzed cell biodistribution, histological changes, and parameters of the immune response. This study aims to evaluate the efficacy of P-MSC and EV therapy in a preclinical murine model in order to contribute to the growing body of knowledge on MSC-based therapies and generate valuable insights into their comparable efficacy in mitigating the detrimental effects of ARDS in vivo across several time points.

## 2. Materials and Methods

### 2.1. Animals

Male and female C57BL/6 mice aged 10–12 weeks and weighing 20–25 g were used for the experiment, and they were purchased from the Lithuanian University of Health Sciences (LUHS) vivarium (veterinary approval no. LT-59-101). Mice were housed in specific pathogen-free cages and maintained at 20–25 °C and 50–70% relative humidity. The Lithuanian State Food and Veterinary Service (approval no. G2-223) approved the experimental protocols. All animal experiments were carried out in the LUHS Biological Research Center (veterinary approval number LT 61-19-004).

### 2.2. Donor Eligibility

Placenta was collected from healthy volunteers undergoing a cesarean section (n = 2) after obtaining written informed consent. The procedure was carried out in accordance with the Lithuanian Bioethics Committee’s standards (approval no. BE-2-105). The eligibility of placental donors was confirmed as per current European Pharmacopoeia standards. After obtaining written informed consent for tissue donation, a questionnaire was performed to determine the risk of blood-borne infections, comorbidities, and consumed medicines. Blood serum testing yielded negative results for the following viral diseases: hepatitis B and C, human immunodeficiency virus I and II, and T. pallidum.

### 2.3. Human Placental Mesenchymal Stem Cell Isolation and Culture

The placenta was processed in clean room facilities using a rigorous quality control strategy that monitors personal and environmental in-process controls for microbiological and viral factors, cell count, and viability in accordance with local legislation requirements. Good Manufacturing Practice (GMP) guidelines were adhered to during the isolation and expansion of human placental mesenchymal stem cells (P-MSCs). To extract P-MSCs, the amniotic and chorionic layers and the umbilical cord were discarded, and the remaining placental tissue was sliced into 2–4 mm pieces and washed in Dulbecco’s phosphate-buffered saline (DPBS, Gibco, Grand Island, NY, USA). The tissue was digested for two hours at 37 degrees Celsius in 0.1% collagenase (NB6, Nordmark, Uetersen, Germany) and DNase solution (Pierce nuclease, Thermo Fisher, Waltham, MA, USA). Following ACK lysis (Gibco, Grand Island, NY, USA), the suspension was filtered and washed. The cells were resuspended in DMEM containing 10% fetal bovine serum (Gibco, Grand Island, NY, USA) and plated in a tissue culture flask (Cell Factories, Corning, Cornig, NY, USA) before being incubated at 37 °C in a humidified atmosphere containing 5% CO_2_. After the initial two days, the medium was replaced every three days. After 8 to 10 days, sub-confluent cells were collected using TrypLE Select (Thermo Fisher, Waltham, MA, USA) and reseeded. This study utilized P-MSCs from passage 4.

### 2.4. Quality Controls

All reagents and materials used for the isolation, expansion, and cryopreservation of human placental stem cells (P-MSCs) were validated as being of clinical quality. To ensure traceability, the lot and/or serial numbers of all reagents and supplies used in each assay were documented. A Neubauer chamber hemocytometer was used to determine cell count, and Trypan Blue was used for viability assessments. During each growth cycle, the number of population doublings (PDs) was calculated as (ln(2)×culture time (days))/(ln((cell harvest number)/(initial cell number seeded))). The cumulative PD (cPD) was calculated by adding passage-specific PDs. The doubling time (PDT) was determined by dividing the number of days required for full cell development by the cumulative PD. To evaluate the sterility of the procedure, aerobic, anaerobic, and fungal bacteria were detected in the cell supernatant. The EP 2.6.27-compliant sterility testing of the final P-MSC preparations was performed using an automated system (BACTECM, BD, Franklin Lakes, NJ, USA). Throughout the process, conditioned medium was included as sterility reference samples. For mycoplasma contaminations, a validated real-time PCR (qPCR) TaqMan^®^-based qPCR Assay (Venor^®^GeM qEP, Minerva Biolabs, GmbH, Berlin, Germany) was used in accordance with EP 2.6.14 on cell supernatants and 1 × 10^6^ cells. Testing for mycoplasma was carried out at each passage, before freezing and after thawing. A kinetic chromogenic quantitative Limulus Amoebocyte Lysate test was performed for endotoxin analysis using an automated spectrophotometer in compliance with EP 2.6.14 (Endosafe NexGen-PTS, Charles River Laboratories, Charleston, SC, USA). Throughout the preparation process, conditioned media-containing cells were added as endotoxin and mycoplasma reference samples. Chromosomes of P-MSCs from the second and fourth passages were isolated at the metaphase stage after cultivating cells to 80% confluence. G-binding staining was applied to prepared metaphase slides, from which twenty metaphases were subsequently analyzed in alignment with the CAT of the EMA guidelines. To induce osteogenic differentiation, a density of 5 × 10^3^ cells/cm^2^ was used to seed the cells in 12-well plates. They were cultured in a Stem Pro Osteogenesis Differentiation medium (Thermo Fisher, Waltham, MA, USA) for 3 weeks. The medium was changed every three days, and the onset of osteoblast formation was assessed by analyzing the expression of Alizarine Red. Next, the cells were treated with the StemPro Adipogenic Differentiation Kit (Thermo Fisher, Waltham, MA, USA) to induce adipogenic differentiation for 2 weeks. Medium changes were carried out every three days, and the presence of adipocytes was evaluated using an Oil Red O (Millipore Sigma, Darmstadt, Germany) stain. Chondrogenic micromass was generated from cultures by seeding 5 μL droplets of cell solution in the center of a 6-well plate. After 14 days in culture, the media were removed from the culture vessel, fixed with formaldehyde, rinsed, and stained with Alcian Blue to evaluate chondrogenic differentiation.

### 2.5. Immunophenotypic Analysis

In this study, 1 × 10^6^ P-MSCs were washed with a FACS buffer comprising DPBS supplemented with 1% bovine serum albumin for immunophenotypic characterization (BSA, Sigma). The cells were then treated for 30 min with the following fluorochrome-conjugated primary antibodies in the dark and at room temperature: anti-CD31-APC, anti-CD45-FITC, anti-CD73-PE, anti-CD90-PE, and anti-HLA-DR-FITC (FITC, BD Biosciences, Franklin Lakes, NJ, USA). The cells were then washed with the FACS buffer before being acquired with a fluorescence-activated cell sorter (BD FACSAria III). BD FACS Diva 8.0.1 was then used to analyze the relative fluorescence intensity of the cells (BD Immunocytometry Systems, Franklin Lakes, NJ, USA).

### 2.6. Collection, Concentration and Analysis of Extracellular Vesicles

The medium with passage 4 cells was collected, centrifuged for 15 min at 2000× *g* at 4 °C, and filtered through a 0.22 µm mPES filter. The EVs were concentrated using a 500 kDa tangential flow filtration filter (Sartorius AG, Gottingen, Germany) in the KrosFlo KR2i system (Repligen Corporation, Waltham, MA, USA). A flow rate of 36 mL/min. with a transmembrane pressure of 3 psi was used, and the medium was concentrated 30 times, diafiltrated with 5 volumes of DPBS, sterilized through a 0.22 µm mPES filter, and stored at −80 °C until further analysis or use in vivo.

The size distribution of particles was analyzed using a Nanosight NS300 instrument (Malvern Panalytical, Malvern, UK) configured with a 488 nm laser. Videos were analyzed using the in-built NanoSight Software NTA 3.2 Dev Build 3.2.16. The camera type, camera level, and threshold were sCMOS, 13, and 2, respectively. A syringe pump with constant flow injection was used. The number of completed tracks in NTA measurements was 3 (a 60 s movie was registered for each measurement). All samples were diluted in PBS to a final volume of 1 mL. The ideal optimal concentration was assessed by pretesting the optimal particle per frame value (20–100 particles per frame). P-MSC EVs attached to aldehyde/sulfate latex beads were labeled with appropriate fluorescent antibodies for CD9, CD63, and CD81 and detected using flow cytometry. The EV suspension was fixed with paraformaldehyde to a 2% final concentration and applied onto formvar/carbon-coated copper grids. The grids were rinsed twice with PBS before incubation with 1% glutaraldehyde and stained negatively with 2% uranyl acetate. Dried samples were examined using a transmission electron microscope (JEOL, USA, Inc., Peabody, MA, USA) equipped with an 11 Megapixel MORADA G2 camera (Olympus Soft Imaging Solutions GmbH, Münster, Germany).

### 2.7. Acute Respiratory Distress Syndrome Induction and Treatment

Mice were weighed and anesthetized using an intraperitoneal injection of ketamine (120 mg/kg) and xylazine (16 mg/kg). Following anesthesia, the mice were positioned supine on a heating mat set to 37 °C. A drop of 2 mg/mL LPS (from *E. coli* O55:B5, Sigma-Aldrich, Darmstadt, Germany) solution in DPBS was applied to each nostril. This administration was repeated until the desired dose (5 μg/g of mouse weight) was achieved. The mice were kept in the supine position under anesthesia for another 30 min, and anesthesia was then reversed using atipamezole (30 mg/kg). At the four-hour mark after induction, mice were intraperitoneally injected with one of the following: 100 µL of a 2 × 10^5^ P-MSC suspension (P-MSC group), a 5 × 10^5^ cell-equivalent P-MSC EV suspension (EV group), or DPBS (control). The mice were categorized into three observation points: 24 h, 72 h, and 168 h post-induction. The experiment was repeated two times using P-MSC and EV from a different donor each time, resulting in the following sample sizes: 24 h control n = 6; 24 h P-MSC n = 8; 24 h EV n = 6; 72 h control n = 6; 72 h P-MSC n = 6; 72 h EV n = 5; 168 h control n = 6; 168 h P-MSC n = 6; 168 h EV n = 6. Healthy mice (n = 6) were used for reference.

### 2.8. Bronchoalveolar Lavage Fluid Collection and Tissue Harvesting

The mice were euthanized via atlanto-occipital dislocation under anesthesia after the allotted period of time. The thorax was incised, and the left bronchus was clipped off using a small hemostatic clip (Weck^®^ Horizon™, Teleflex Incorporated, Wayne, PA, USA) to prevent the lavage fluid from entering the left lung and disrupting the tissue that would be used for histology. Afterwards, an incision in the neck skin near the trachea was made, and the trachea was separated from the surrounding tissue. A small incision was made in the anterior part of the trachea. A 22 g catheter was inserted 5 mm into the trachea and stabilized by tying the trachea around the catheter with a suture. Then, 0.5 mL of DPBS was gently instilled through the catheter while observing the right lung for inflation, held for 60 s, and aspirated. This was repeated a total of three times, using fresh DPBS each time. The collected bronchoalveolar lavage fluid (BALF) was pooled and placed on ice until further processing (for no longer than 2 h). The right lung was removed and divided into two pieces: One was stored in RNAlater for RT-qPCR at −20 °C, and the other was stored at −20 °C for biodistribution analysis. The left lung was removed, fixed in 10% formalin for 24 h, and embedded in paraffin for histological and immunohistochemical examination. The liver, spleen, and kidneys were removed and stored at −20 °C for the biodistribution study.

### 2.9. Bronchoalveolar Lavage Fluid Analysis

The bronchoalveolar lavage (BALF) volume was measured, and the BALF was centrifuged at 400× *g* for 10 min at 4 °C. The supernatant was aspirated; Halt™ Protease Inhibitor Cocktail (Thermo Scientific, Walham, MA, USA) was added, and it was frozen at −80 °C. The cell pellet was resuspended in 200 μL of ACK lysis buffer and incubated for 2 min at room temperature. Then, 1 mL of Hanks’ Balanced Salt Solution (HBSS) (-Ca, -Mg) was added, and the mixture was centrifuged at 400× *g* for 10 min. at 4 °C. The supernatant was removed, and the cells were resuspended in 200 μL of HBSS and counted with a hemocytometer to enumerate the absolute cell count. Finally, after repeated centrifugation, the cell pellet was resuspended in 5 μL of HBSS. The entire volume was smeared onto a microscope slide. The smear was air-dried for 1 min, fixed with methanol for 1 min, and stained with Wright–Giemsa stain. Two hundred cells for each sample were counted to enumerate the relative numbers of neutrophils, macrophages, and lymphocytes. The supernatant’s total protein concentration was determined using the Pierce™ BCA Protein Assay (Thermo Scientific, Walham, MA, USA) according to the manufacturer’s instructions.

### 2.10. Histology and Immunohistochemistry

The study material was fixed in a 10% buffered formalin solution and embedded in paraffin using standard methodology. Sections of 3 μm thickness were prepared using a rotary microtome and then stained with hematoxylin-eosin and pikro-Mallory. The slides were scanned with a 3D Histech Pannoramic MIDI scanner and analyzed using the 3D Histech Pannoramic Viewer and Histoquant software (QuantCenter version 2.3, 3DHISTECH, Budapest, Hungary). Twenty 40× amplification fields were graded in accordance with a lung injury scoring system, as described by Aeffner et al. [30] (Appendix A).

For immunohistochemical analysis, 3 μm sections were placed on Super Frost Plus slides (Sigma-Aldrich, Darmstadt, Germany). The slides were kept at 58 °C in a temperature-controlled chamber for 3 h prior to staining. Deparaffinization was performed using xylene and ethanol on a Varistain Gemini staining automaton. The sections were washed with distilled water. The epitope was retrieved using an RHS-1 microwave unit (Milestone Medical, Bergamo, Italy) by incubating the sections in a TRIS/EDTA pH 9.0 buffer at 110 °C for 8 min. Further immunohistochemical staining was performed using Shandon Coverplate slides. After blocking endogenous peroxidase, the sections were incubated with primary antibodies against myeloperoxidase (MPO, rabbit monoclonal, clone EPR20257, Abcam, ab208670, Cambridge, UK) for 1 h. The sections were then processed using the Agilent Dako EnVision+ single reagent (HRP rabbit, Agilent Technologies, Santa Clara, CA, USA) visualization system according to the manufacturer’s recommended protocol. Recommended control tissue sections were used for positive immunohistochemical analysis controls, and negative controls were performed on the same sections using immunoglobulins of the appropriate isotype instead of the primary antibody.

### 2.11. Immunohistochemistry Quantification

Slides were analyzed using QuPath version 0.4.3 [31]. Each slide was manually annotated with ten non-overlapping 500 μm diameter circle annotations distributed throughout the lung section in areas with the most visible MPO-positive cell infiltration. When annotating, areas with larger airways or vessels (>80 μm) and scanning artifacts were excluded (see Appendix A for annotation examples). The positive cell detection function was used to mark positively stained cells, and the parameters were fine-tuned to ensure the best possible cell selection (see Appendix A for examples). Cell detection accuracy was confirmed in a range of different samples, and annotations were placed by an experienced pathologist (D.P.) who was blinded to the intervention used. The number of positive cells per mm^2^ is used for statistical analysis. The annotations for each sample were not averaged before statistical analysis but instead grouped by intervention and time point and analyzed individually.

### 2.12. Quantitative Real-Time Polymerase Chain Reaction

Mice lung tissue was removed from RNAlater, ground in liquid nitrogen, and homogenized with a 20G syringe in the RNA isolation lysis buffer (Invitrogen, Carlsbad, CA, USA). The total RNA was isolated using the PureLink™ RNA Mini Kit (Invitrogen, USA), and reverse transcription was performed with the High-Capacity RNA-to-cDNA™ Kit (Applied Biosystems, Waltham, MA, USA) according to the manufacturer’s instructions. The gene expression of target genes (*Il1b* (assay Mm00434228_m1), *Tnf* (assay Mm00443258_m1), *Il17a* (Mm00439618_m1), and *Foxp3* (assay Mm00475164_m1)) was assessed using the quantitative real-time polymerase chain reaction (qRT-PCR) method. Gene expression analysis was performed using the QuantStudio 5 real-time thermocycler (Applied Biosystems, Waltham, MA, USA) in MicroAmp Optical 96-well plates (Applied Biosystems, Waltham, MA, USA). TaqMan Universal Master Mix II with the UNG 2 × (Applied Biosystems, Waltham, MA, USA) reaction mixture and TaqMan Gene Expression Assay primer sets (Applied Biosystems, Waltham, MA, USA) were used to specifically amplify target gene fragments. Reference genes beta-actin (ACTB; assay Mm00607939_s1), glyceraldehyde 3-phosphate dehydrogenase (GAPDH; assay Mm99999915_g1), and peptidylprolyl isomerase A (PPIA; assay Mm02342430_g1) were tested in order to be used as endogenous controls for result normalization. After evaluating the expression levels across samples from different groups, ACTB was chosen as the endogenous control since it displayed the most stable expression levels, which is consistent with the findings of Fragoulis et al. [32]. Three technical replicates were performed for each sample. qRT-PCR was performed in the thermocycler with the following temperature settings: 50 °C for 2 min, 95 °C for 10 min, and 40 cycles of 95 °C for 15 s, and 60 °C for 1 min. Relative gene expression was calculated using the ΔΔCt method [33]. Gene expression fold changes (FCs) are reported relative to healthy mouse gene expression levels.

### 2.13. Luminex and ELISA

BAL supernatants were analyzed using a custom Luminex panel, including IFN-γ, IL-6, IL-10, and IL-12p40 (R&D Systems, Minneapolis, MN, USA), according to the manufacturer’s instructions. The lower detection limits were as follows (in pg/mL): 1.34 for IFN-γ, 3.03 for IL-6, 0.99 for IL-10, and 7.81 for IL-12p40. We addressed values below the detection limit using a substitution approach. Specifically, when an analyte was not detected, the value was replaced with a value equal to half of the lower limit of detection. By utilizing this substitution method, we aimed to prevent the complete exclusion of values that could contribute meaningful information to our analysis. Where possible, Luminex assay values below detection limits were extrapolated.

### 2.14. Biodistribution Assay

MSC biodistribution was evaluated by human DNA (hDNA) detection in murine organs, as described by Creane et al. [34]. Briefly, liver, kidney, lung, and spleen tissues were minced by cutting and weighed, and 25 mg of tissue was used for DNA isolation as per the manufacturer’s instructions (PureLink Genomic DNA Mini Kit, Invitrogen, USA) using aseptic technique and sterile instruments. Additionally, DNA was isolated from 5 × 10^5^ MSC cells. Control mice tissue was used as the negative control. A serial dilution of MSC DNA (0–200 ng, 8 standards) was prepared prior to PCR for standard curve generation. To detect hDNA, a primer-probe-based qPCR assay targeting the human-specific sequence of the Alu repeat was used. Alu-qPCR was carried out with primers (forward primer (101 F), 5′-GGTGAAACCCCGTCTCTACT-3′; reverse primer (206 R), 5′-GGTTCAAGCGATTCTCCTGC-3′) and a hydrolysis probe ((144RH) 5′-CGCCCGGCTAATTTTTGTAT-3′) specifically designed and described in the Funakoshi et al. [35] study. Alu-qPCR samples were incubated at 50 °C for 2 min and 95 °C for 10 min, followed by 40 cycles of 95 °C for 15 s and 60 °C for 1 min. Three technical replicates were performed for each sample. Log10 standard MSC DNA concentrations were plotted against the equivalent Ct values. Human DNA concentrations in mice tissue samples were calculated using the standard curve. Cell equivalent numbers were calculated as described by Creane et al. [34]. Finally, human cell equivalents per 25 mg of organ mass were calculated.

### 2.15. Statistical Analysis

Results are described as medians with respect to the 25–75th interquartile range (IQR). Statistical analysis was performed using GraphPad Prism version 10.0.0 (GraphPad Software, San Diego, CA, USA), with a significance level of *p* < 0.05. Differences across groups were evaluated with the Kruskal–Wallis test, and Dunn’s test was used for pairwise comparisons. Healthy mice sample results are described in Appendix A. All control, P-MSC, and EV results and statistics are outlined in Appendix A.

## 3. Results

### 3.1. Human Placental Cells Display Mesenchymal Stem Cell Characteristics

P-MSCs displayed a predominant round-spindle shape, which is consistent with the classical mesenchymal stem cell shape (Figure 1A). Up until passage 6, there were no major modifications in cell morphology. Furthermore, P-MSCs displayed multilineage differentiation potential in adipogenic, chondrogenic, and osteogenic cultures (Appendix A). The G-banding technique analyzed P-MSC chromosomes at the second and fourth passages. Following the Giemsa staining of metaphase slides, both stages of the product displayed a regular karyogram (46, XX), suggesting the maintained chromosomal integrity of P-MSCs during drug development (Figure 1B). P-MSCs had an exceptionally high viability of 95.83 ± 1.94%, which was assessed in each passage after detachment (Figure 1C). Population doubling times at P0 were lower than in the other passage cells (*p* < 0.0001). P1 and P2 showed no significant differences, but P3, P4, and P5 were significantly shorter than P2 or P3 (*p* < 0.0001, Figure 1D). The cumulative generations between each passage were significantly different from the previous passage (Figure 1E).

No bacterial growth was observed in the cell supernatant throughout the process, and no tested samples were contaminated with mycoplasma. The results of the kinetic chromogenic endotoxin analysis showed that the endotoxin levels were below the acceptable limit of 5 EU/kg, which indicates that the P-MSC preparations were not contaminated with endotoxins. The results of immunophenotypic characterization showed that more than 95 percent of P-MSCs expressed CD73, CD90, and CD105, while less than 1 percent expressed CD31, CD45, and HLA-DR (Figure 1F). The in vitro expansion of P-MSCs did not affect the expression profile of these markers.

### 3.2. Human Placental Mesenchymal Stem Cells Produce Exosomes Displaying Characteristic Features

P-MSC-secreted EVs displayed typical exosome size, morphology, and surface marker characteristics. Transmission electron microscopy (TEM) visualization showed typical exosome morphology in all samples (Figure 2A,B). In the 30× concentrated EV samples from fourth passage cells, the mean EV size was 133 nm and the mode size was 87.2 nm, with a concentration of 1.26 × 10^11^ particles/mL (Figure 2C). When the total particle number was divided by the cell number at harvest, the median number was 2.6 × 10^4^ particles per cell. P-MSC EVs in bead-based flow cytometry showed the following median (min–max) percentages of exosomal markers: CD63 18.1 (17.3–18.9)%, CD9 33.3 (31.2–35.4)%, and CD81 30.7 (28.0–33.4)%.

### 3.3. Extracellular Vesicle Therapy Significantly Reduces Lung Injury Scores

To evaluate the therapeutic effects of P-MSC or their EV, we instilled LPS intranasally in mice and, 4 h later, injected them intraperitoneally with PBS (control), P-MSC, or EV. We then analyzed the parameters of lung injury at 24, 72, and 168 h after induction. All mice in the treatment groups survived, whereas one mouse in the control 24 h group died. Each EV-treated mouse received a dose of 1.2–1.3 × 10^10^ particles. LPS instillation resulted in significant neutrophil infiltration into the alveolar and interstitial spaces, alveolar septal thickening, areas of hemorrhage, and atelectasis 24 and 72 h after injury (Figure 3’s panels). There was some resolution of these changes 168 h after the injury. Histological lung injury score grading (Figure 3 graph) revealed that EV therapy significantly (*p* = 0.006) reduced the lung injury score 168 h after injury, and the difference between P-MSC and EV therapies almost reached significance (*p* = 0.05).

### 3.4. Myeloperoxidase Immunohistochemistry Reveals Neutrophil Infiltration Reduced by Human Placental Stem Cells and Their Extracellular Vesicles

LPS instillation induced the significant infiltration of myeloperoxidase (MPO)-positive cells in the lung tissue (Figure 4 panels). MPO-positive cells (number of cells positive per mm^2^) were quantified using QuPath, which allowed for the statistical analysis of the results. Since MPO is mostly abundant in neutrophils [36], this immunohistochemical analysis allows for a more objective and precise quantification of neutrophil infiltration. In the control group, 24 and 72 h after induction, there was a significant recruitment of neutrophils (Figure 4 graph), when compared to samples from healthy mice. Afterwards, 168 h after injury, the number of infiltrating neutrophils returned to baseline levels. Twenty-four hours after injury, P-MSC- or EV-treated mice both displayed significantly lower levels of MPO-positive cells when compared to the control group. Seventy-two hours after injury, only the P-MSC group displayed a statistically significant reduction in MPO-positive cell count, possibly due to high variability across samples at this time point. Interestingly, there were significantly fewer MPO-positive cells in the P-MSC group than in the EV group. Overall, LPS instillation resulted in acute infiltration with respect to MPO-positive cells, which was reduced by MSC-based therapy. Furthermore, the quantification of IHC staining allowed for a robust and reproducible evaluation of positive cell counts, with more sensitivity for subtler differences in inflammation intensity amongst treatment groups.

### 3.5. Human Placental Stem Cell and Extracellular Vesicle Treatment Reduce Cell Count and Protein Concentration in Bronchoalveolar Lavage Fluid and Modulate Cytokine Profiles

Bronchoalveolar lavage fluid (BALF) cell count changes are important indicators of the inflammatory response to lung injury [30]. Following LPS instillation, there was a numerical increase in total cell count compared to the baseline at 24 h in all three groups, and this was primarily attributed to neutrophil infiltration (Figure 5). These changes in total cell counts and relative neutrophil counts returned to levels observed in healthy mice 168 h after injury in all three groups. As for the macrophage population, there was a significant decrease from baseline in the 24 and 72 h groups, which could be explained by the large influx of neutrophils. When comparing the control group to the treatment groups, it was observed that the EV treatment group exhibited a significantly reduced number of cells 72 h post-injury. While the P-MSC group did not reach statistical significance (*p* = 0.06), it showed a clear trend towards a reduction in the total cell count. Taken together, these findings strongly suggest that at this dose, EV treatment expedites the resolution of cell infiltration subsequent to LPS instillation more effectively than P-MSC.

Loss of alveolar barrier integrity is an important pathophysiological change in ARDS, for which an increase in total BALF protein concentration is a highly relevant indicator [30]. In the control group, 24 h after lung injury, there was a significant increase in protein concentration, which peaked at 72 h and almost returned to the baseline at 168 h. Treatment with EV displayed a numerical decrease in alveolar permeability surrogate (total protein) 24 h after injury; however, it did not reach statistical significance (*p* = 0.15). This decrease from the control in the surrogate of alveolar permeability was even more pronounced and statistically significant 72 h after injury in both treatment groups (Figure 6). Therefore, both P-MSC and EV treatments accelerate the resolution of alveolar barrier integrity.

The analysis of cytokine levels in BALF revealed significant insights into the immune response following LPS-induced lung injury. While protein concentration analysis often benefits from normalization relative to the total protein content, in this context, total protein concentration is inappropriate and might be a misleading reference point since the total protein levels vary significantly due to differing levels of alveolar barrier injury and inflammation across groups [37]. Therefore, raw Luminex protein values are presented in Figure 6. IL-6, which is a highly relevant pleiotropic cytokine, was significantly elevated 24 h after injury in all groups and returned to baseline at 168 h. There were no statistically significant differences in IL-6 levels across all intervention groups. As for IL-10, which is a potent anti-inflammatory cytokine that is usually secreted by Treg lymphocytes [38], there was no statistically significant elevation of IL-10 levels from the healthy baseline in all three groups; however, when comparing therapy and control groups, there were significantly higher IL-10 levels in the 72 h EV group (*p* = 0.008) and an increase in the P-MSC 72 h group, which did not reach statistical significance (*p* = 0.08). As for IL-12p70, which is tightly linked with the differentiation of IFN-γ-producing lymphocytes [39], there was a significant elevation from the baseline in the control and P-MSC groups 72 h after injury, whereas the 72 h EV group remained at baseline levels. IFN-γ levels were also greatly increased 24 h and 72 h after injury, without any significant differences between the intervention groups. Overall, LPS induced a robust immune response with the strong activation of the Th1 response, which was not necessarily blunted by P-MSC or EV therapy; however, in the EV group, a much stronger anti-inflammatory signal was observed, which could at least partly explain the therapeutic effect outlined in the previous sections.

### 3.6. Intraperitoneal Human Placental Stem Cell Administration Results in Negligible Liver, Kidney, Spleen, and Lung Cell Engraftment

After the intraperitoneal injection of P-MSC, we evaluated the biodistribution of cells with qRT-PCR by quantifying the amount of human DNA (hDNA) detected in the kidneys, liver, spleen, and lungs. Our analysis revealed that there was no hDNA detected in the liver across all time points. There was an insignificant amount of hDNA in the kidneys and spleen in some of the 24 h time point samples (median [IQR] of 0.15 [0–0.20] and 0.14 [0–0.18] cell equivalent/25 mg tissue, respectively). We detected no hDNA at the 72 h time point across all samples. Interestingly, at the 168 h time point, hDNA was detected only in the lung samples (median [IQR] of 0.09 [0.00–0.27] cell equivalent/25 mg tissue). No hDNA was detected in samples from control mice. Overall, these are negligible cell equivalent numbers, considering that each mouse in the P-MSC treatment group received an injection of 2 × 10^5^ cells.

### 3.7. Human Placental Stem Cells and Their Extracellular Vesicles Significantly Reduce Inflammatory Cytokine Gene Expression

Lung tissue gene expression analysis (shown in Figure 7) in control group mice revealed that 24 h after injury, the expression levels of *Il1b*, *Tnf*, and *Il17a* genes increased significantly from the baseline in all three groups, and *Il1b* and *Tnf* returned to values seen in healthy mice 168 h later (Figure 7). However, *Il17a* expression levels remained elevated 168 h after the injury. There were no statistically significant changes in *Foxp3* gene expression (Appendix A), a crucial regulatory gene that plays a central role in the development and function of regulatory T cells [40]. These results suggest a robust inflammatory response in the injured lung tissue during the acute period, which mostly resolved after 168 h. The sustained elevation in *Il17a* expression could indicate the prolonged involvement of the Th17 immune axis. As for cell-based treatment interventions, the most significant differences when compared with the control group were observed 72 h after injury. Notably, both P-MSC and EV therapy groups all showed reduced expression levels relative to the *Il1b*, *Tnf*, and *Il17a* genes. EV further reduced the expression level of *Il17a* to baseline levels 168 h after injury. Overall, these data would suggest that P-MSC and their EV accelerate the suppression of pro-inflammatory gene expression in lung tissue.

## 4. Discussion

The present study sought to investigate the therapeutic potential of human placental mesenchymal stem cells (P-MSCs) and their EVs in the context of acute respiratory distress syndrome (ARDS). To our knowledge, this is the first animal study that investigates placental mesenchymal stem cell and extracellular vesicle therapy effects at multiple time points. Our results reveal several critical findings related to the characteristics of P-MSCs, the biodistribution of administered cells, the impact on lung injury, and the modulation of the inflammatory response.

The placenta is an excellent source of mesenchymal stem cells. Firstly, it is readily available and not associated with ethical concerns, as it is typically discarded after childbirth. Furthermore, placental cells are young perinatal cells, which reduce the risk of cellular senescence. As for our findings, P-MSC exhibited classical mesenchymal stem cell characteristics: adherence to plastic, multilineage differentiation, typical surface markers along with chromosomal integrity, high viability, and proliferation capacity. These characteristics are in line with previous reports [41,42]. Additionally, our data on EV characterization confirmed that P-MSC-derived EVs displayed typical exosome features, including size, morphology, and marker expression [43]. All in all, these features make P-MSC an appealing choice for both MSC therapy and exosome production.

Intranasal LPS administration successfully induced acute lung injury in mice that displayed the necessary typical features of histological lung injury evidence, alteration of the alveolar–capillary barrier, and clear evidence of an inflammatory response [30]. Furthermore, the higher levels of IL-1β, IL6, IL-12p70, IL-17A, TNF-α, and IFN-γ are consistent with the inflammatory profile of COVID-19-associated lung injury [44]. We then demonstrated that both P-MSC and EV therapies reduced the degree of lung injury by accelerating recovery. Histological lung injury grading revealed a significant injury reduction only in the EV group. However, IHC data proved to be more sensitive, since there was a decreased MPO-positive cell count in the P-MSC and EV groups 24 h after injury and 72 h after injury in the P-MSC group. As for the BALF cell count, EV therapy significantly reduced the total cell count 72 h after injury, whereas the P-MSC reduction did not reach statistical significance. Also, 72 h after injury, we report a decrease in total BALF proteins in both treatment groups, which indicates a resolution of the alveolar–capillary barrier injury. The differences in inflammatory BALF cytokine concentrations were not significant; however, there was an increase in IL-10 levels in the treatment groups, which was more pronounced and statistically significant in the EV group. Finally, 72 h after injury, all tested treatment modalities significantly reduced the expression of the Th1 and Th17 axes’ pro-inflammatory cytokine genes *Il1b*, *Tnf*, and *Il17a*, where EV treatment had a larger effect on *Il17a* gene expression than P-MSC. Overall, at the tested dose, EV induced a stronger anti-inflammatory response than P-MSC and was more effective at reducing the evaluated parameters of lung injury. Furthermore, we observed the most notable differences at the 72 h time point, strongly advocating its prioritization in future pre-clinical ARDS investigations. Other researchers describe similar lung injury reductions after MSC EV applications in murine lung injury models [21,45,46]. Tieu et al. recently published an article outlining some important LPS lung injury model considerations. Firstly, they describe that IL-6 concentrations and BALF cell count changes peak at 10–24 h post-injury, whereas BALF protein concentrations and histological changes peak at 72 h post-injury. This is mostly consistent with our findings, although in our study, the return to baseline levels was slightly longer. Additionally, Tieu et al. found that the optimal time for EV administration was 24 h after injury because it led to the highest EV accumulation in the lungs. This is an important consideration for further studies, which could increase EV efficacy [47]. All in all, our findings are in line with those of other groups, although different doses, cell sources, and administration routes were used.

The biodistribution study revealed that after intraperitoneal P-MSC injection, there was no evidence of significant cell engraftment in the liver, kidney, lungs, or spleen across all time points. The intraperitoneal route was chosen to avoid cell entrapment in the lung’s blood vessels, which can reduce P-MSC viability and lifespan [48] and risk further disrupting the lung’s blood flow [49]. Furthermore, other studies show that intraperitoneal administration is effective in ameliorating lung injury [50], intraperitoneally injected cells aggregate in the surrounding tissue and influence the recipient’s immune system via soluble factors [51], and in some cases, intraperitoneal injection is superior to intravenous injection [52]. All in all, the intraperitoneal injection of P-MSC is effective, and it most likely alleviates inflammation via P-MSC-secreted bioactive molecules; however, further studies are required to evaluate the comparative efficacy of MSC and EV in different routes of administration and dosages.

Several limitations of our study should be acknowledged. First of all, the sample size, while large enough for statistical analysis, was still relatively small. A larger sample size could provide insight into more subtle changes resulting from lung injury and cell therapy. Secondly, no physiological parameters were evaluated, which would have given important data on changes in respiratory function. At this stage, the proof of concept can be observed, and for further translatability to humans, correlations with physiological parameters could be investigated in future studies. Thirdly, we did not evaluate cytokine changes in the serum, which are more often used in clinical practice as biomarkers than BALF cytokines. Blood serum cytokine changes, although crucial in clinical trials, serve as indirect indicators of lung injury. Therefore, they hold secondary significance in assessing the effectiveness of therapies in reducing tissue damage. Additionally, we only evaluated intraperitoneal administration in a single dose; however, alternative doses along with other administration routes could have a different effect on biodistribution and efficacy, with limited value for translatability to human studies. Moreover, due to the limited sample size of placental donors in our study, any potential inter-donor variations could not be assessed. We think that this issue can be addressed most comprehensively by dedicated P-MSC and EV quality studies immediately after proof-of-concept non-clinical studies, as observed in our study. Additionally, we also studied EVs collected from P-MSC cultures grown with regular fetal bovine serum (FBS). While this can lead to significant contamination with fetal serum EV [53], our own experience with placental MSC has shown that serum EV depletion negatively impacts cell proliferation (unpublished data). This is in line with recently published data from other researchers, who have reported that EV-depleted FBS increased the population doubling time and diminished the immunomodulatory effect of umbilical MSC [54]. To add to this, the depletion of extracellular vesicles from fetal bovine serum has been shown to negatively affect gene and miRNA expressions during the proliferation and differentiation of skeletal muscle cells in vitro [55]. Since the goal of this experiment was to evaluate the effect of both placental MSC and their EV in an animal lung injury model, we decided to use regular FBS to avoid negatively impacting the potency of the cells or their EV. Furthermore, while the murine model used in our study shares certain pathophysiological features with ARDS and COVID-19, it still represents a simplified version of human disease. Considering the limited translatability of a murine model, we are planning to test relevant differences directly in a first-in-human pharmacodynamic study. Finally, while we did evaluate different time points, the study covered a relatively short time frame of up to 7 days. While we have not yet investigated the long-term effects and potential complications beyond this time frame, this will not deter us from considering a subsequent non-clinical study and proceeding with human trials.

In conclusion, our findings support the therapeutic potential of P-MSCs and their EVs in lung injury. These findings highlight the capacity of P-MSC and EV to accelerate the resolution of lung injury, as evidenced by reduced histopathological damage, diminished inflammatory markers, and improved alveolar barrier integrity. Furthermore, we show that EVs at the dose used were more effective than P-MSCs in reducing lung injury.

## Figures and Tables

**Figure 1 cells-12-02729-f001:**
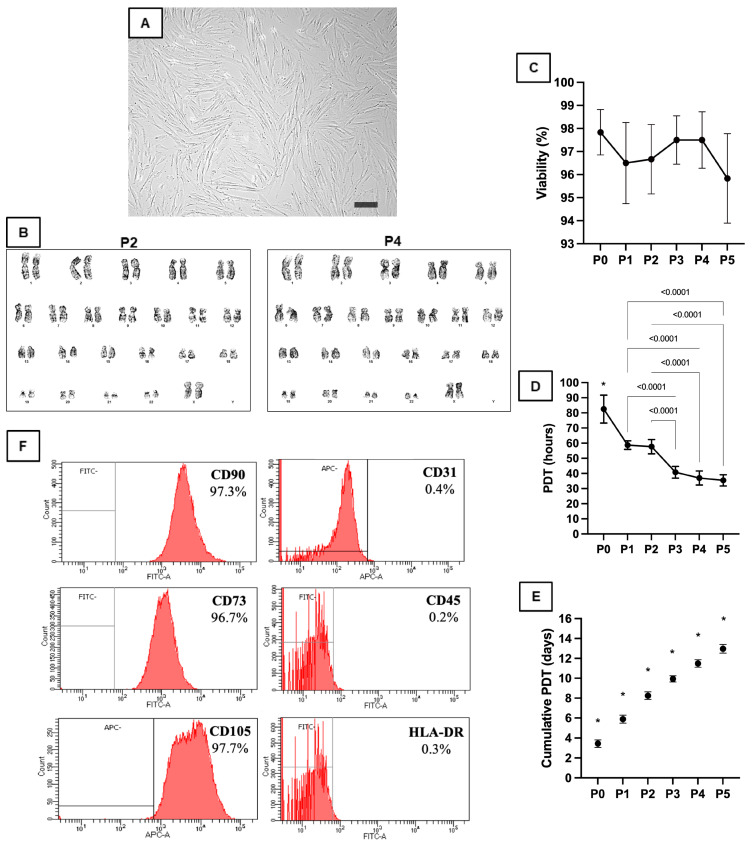
P-MSC characterization: (**A**) P-MSC morphology at passage 4. 80% confluence: elongated, fibroblastic-like, spindle-shaped cells. Original magnification ×100, phase-contrast light microscopy. The scale bar represents 100 µm. (**B**) Karyotyping analysis of P-MSCs at the second (**left**) and fourth passage (**right**) using G-banding. (**C**) Cell culture viability graph. (**D**) Population doubling time graph. (**E**) Cumulative population doubling time graph. *—*p* < 0.05 when compared with the earlier passage. P—passage; PDT—population doubling time. (**F**) Immunophenotypic characterization of cultured human placental stem cells.

**Figure 2 cells-12-02729-f002:**
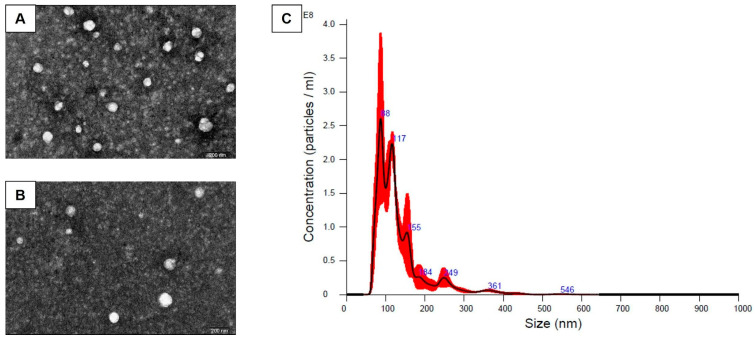
EV characterization results. (**A**,**B**) Representative transmission electron microscopy micrographs of P-MSC EV; scale bar indicates 200 nm. (**C**) Nanoparticle tracking analysis of the total vesicles isolated from the P-MSC-conditioned medium. Line corresponding to average number and size of isolated particles, calculated from the mean of 3 videos lasting 60 s per sample.

**Figure 3 cells-12-02729-f003:**
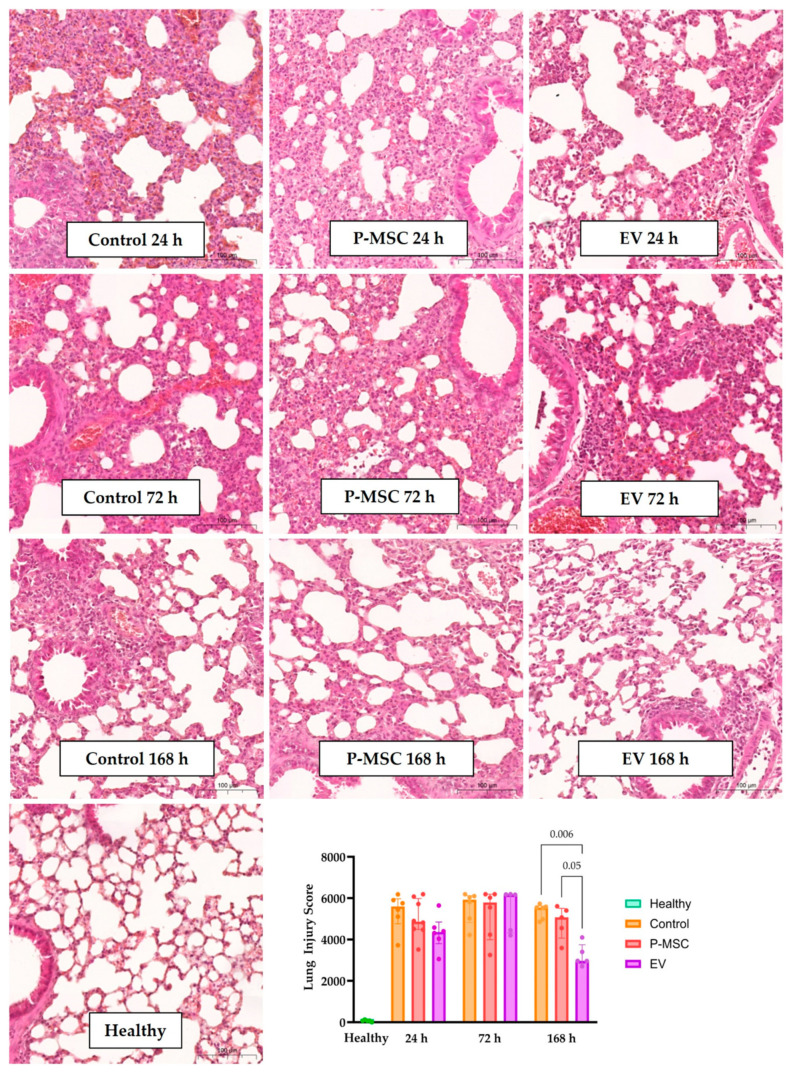
Histological samples of lung sections from all groups (H&E stain) and semi-quantative grading of histological lung injury. Panels show representative images of histological sections. The scale bar indicates 100 µm. The graph displays the results of histological lung injury grading. Individual data points are shown. Significant *p* values are displayed as comparisons across groups at each time point. P-MSC—human placental stem cell-treated; EV—P-MSC-derived EV-treated. Columns represent median values, with 25th and 75th quartiles as error bars.

**Figure 4 cells-12-02729-f004:**
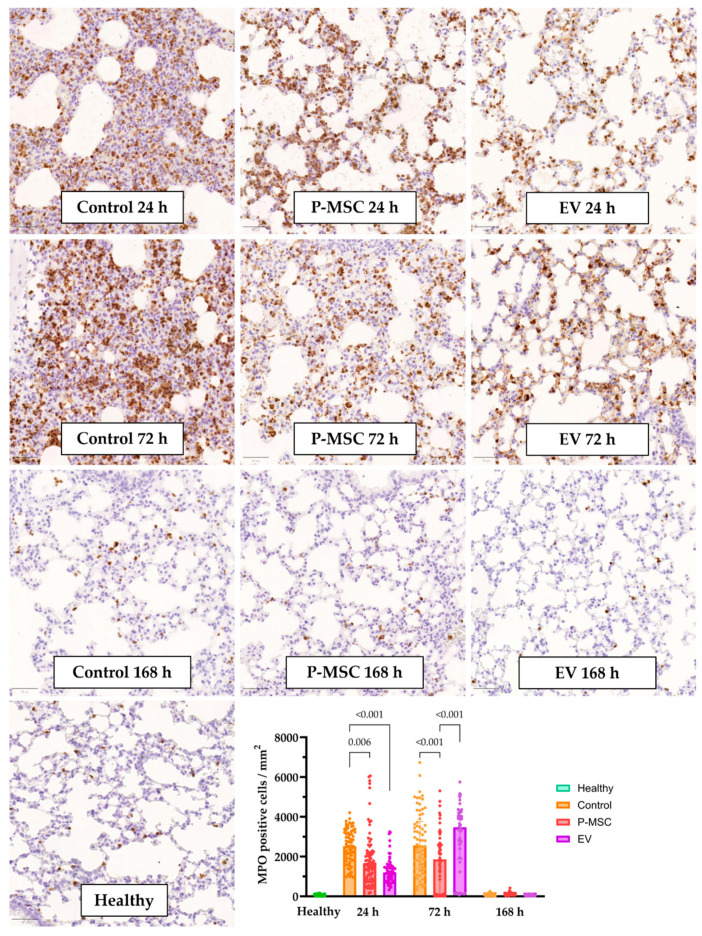
Myeloperoxidase immunohistochemistry samples of lung sections from all groups. The scale bar indicates 50 µm. Panels show representative images of immunohistochemical sections. Graphs display results from the quantitative analysis of MPO-positive cells in lung samples. Individual data points are shown (10 annotations per sample). Significant *p* values are displayed as comparisons across groups at each time point. P-MSC—human placental stem cell-treated; EV—P-MSC-derived EV-treated. Columns represent median values, with the 25th and 75th quartiles as error bars.

**Figure 5 cells-12-02729-f005:**
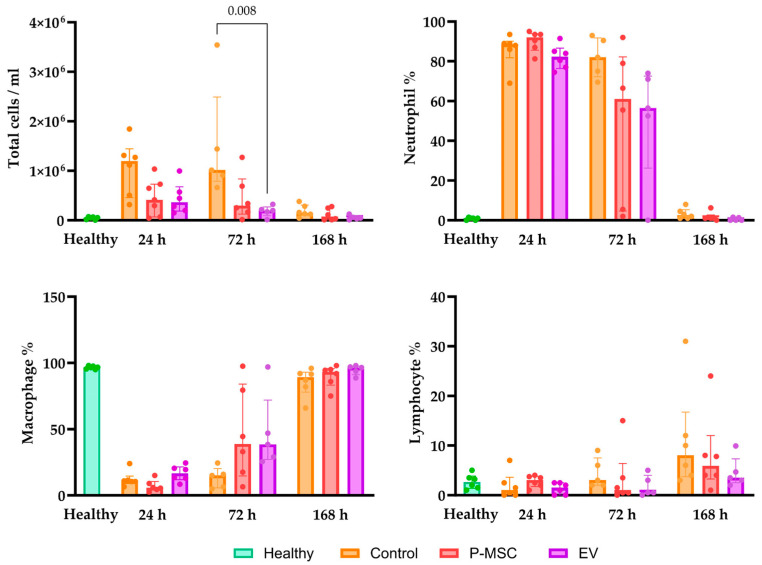
Total and relative cell counts of neutrophils, macrophages, and lymphocytes in bronchoalveolar lavage fluid (BALF). Individual data points are shown. Significant *p* values are displayed as comparisons across groups at each time point. P-MSC—human placental stem cell-treated; EV—P-MSC-derived EV-treated. Columns represent median values, with the 25th and 75th quartiles as error bars.

**Figure 6 cells-12-02729-f006:**
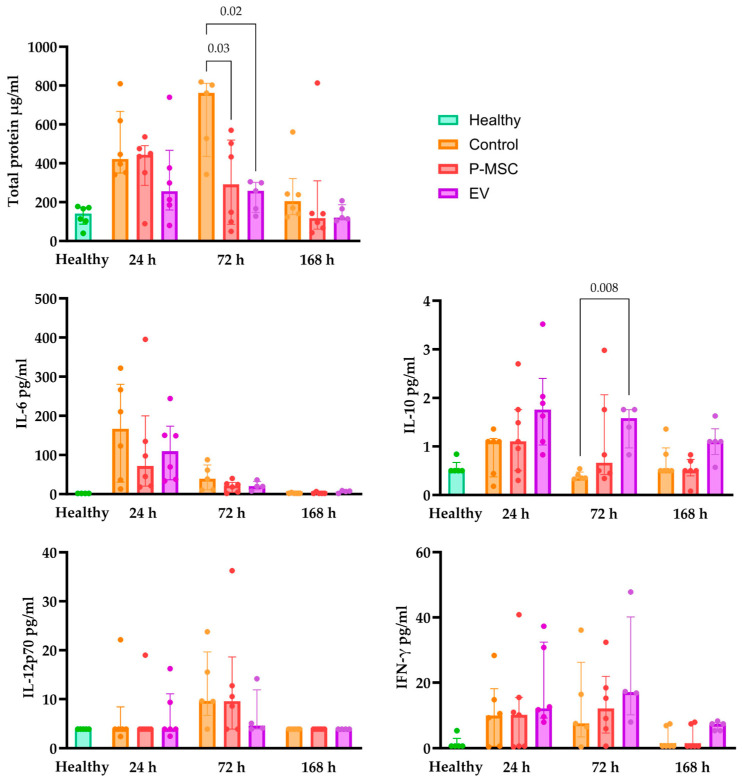
Bronchoalveolar lavage fluid total protein concentration and cytokine concentrations. Individual data points are shown. Significant *p* values are displayed as comparisons across groups at each time point. P-MSC—human placental stem cell-treated; EV—P-MSC-derived EV-treated. Columns represent median values, with the 25th and 75th quartiles as error bars.

**Figure 7 cells-12-02729-f007:**
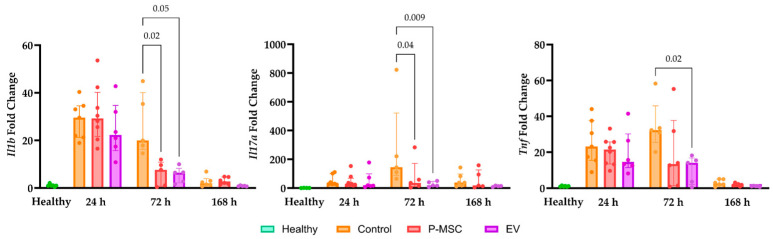
Changes in *Il1b*, *Tnf*, *Il17a*, and *Foxp3* gene expression in lung tissue. Gene expression changes are presented as fold changes in comparison with tissue from healthy mice. Individual data points are shown. Significant *p* values are displayed as comparisons across groups at each time point. P-MSC—human placental stem cell-treated; EV—P-MSC-derived EV-treated. Columns represent median values, with the 25th and 75th quartiles as error bars.

## Data Availability

The datasets used and analyzed during the current study are available from the corresponding author upon request.

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
