# Peer review of "Human Placental Mesenchymal Stem Cells and Derived Extracellular Vesicles Ameliorate Lung Injury in Acute Respiratory Distress Syndrome Murine Model"

_cells, 2023, doi:10.3390/cells12232729_

Round 1

Reviewer 1 Report

Comments and Suggestions for Authors

This study investigates the therapeutic potential of human placental mesenchymal stem cells (P-MSCs) and their extracellular vesicles (EV) in a murine model of acute respiratory distress  syndrome (ARDS). They induced ARDS-like lung injury in mice using intranasal LPS instillation and evaluated histological changes, neutrophil accumulation through immunohistochemistry, bronchoalveolar lavage fluid cell count, total protein and cytokine concentration, as well as lung gene expression changes at three time points: 24, 72, and 168 hours. They found that both P-MSCs and EV treatments reduced histological evidence of lung injury, decreased neutrophil infiltration, and improved alveolar barrier integrity. Analysis of cytokines and gene expression revealed that both treatments accelerated inflammation resolution in lung tissue. However, EV therapy was more effective than P-MSCs in reducing most aspects of lung injury.

Concerns:

How many animals have been used in each group? Only the control group is clear (n=5).

How discern the authors the EV from the cells in culture and the EV from the serum used in the culture medium?

Author Response

Dear Reviewer,

Thank you for your thorough review and valuable comments on our manuscript. We would like to address your concerns:

  • As for the animal numbers, we agree that it should be clearer. We added a more thorough animal sample size description in the 2.7 Methods section: The experiment was repeated two times, using P-MSC and EV from a different donor each time, resulting in the following sample sizes: 24 h control n = 6; 24 h P-MSC n = 8; 24 h EV n = 6; 72 h control n = 6, 72 h P-MSC n = 6, 72 h EV n = 5; 168 h control n = 6; 168 h P-MSC n = 6, 168 h EV n = 6. Healthy mice (n = 6) were used for reference. We also added individual data points to our graphs.
  • As for serum EVs, we have used regular fetal bovine serum (FBS) in our culture, therefore, we cannot discern exactly which EV are from MSC, and which are from FBS. We have included in the discussion our reasoning behind using FBS: Additionally, we also studied EVs collected from P-MSC cultures grown with regular fetal bovine serum (FBS). While this can lead to significant contamination with fetal serum EV [56], our own experience with placental MSC has shown that serum EV depletion negatively impacts cell proliferation (unpublished data). This is in line with recently published data from other researchers, who have reported that EV-depleted FBS increased the population doubling time and diminished the immunomodulatory effect of umbilical MSC [57]. To add to this, depletion of extracellular vesicles from fetal bovine serum has been shown to negatively affect gene and miRNA expressions during proliferation and differentiation of skeletal muscle cells in vitro [58]. Since the goal of this experiment was to evaluate the effect of both placental MSC and their EV in an animal lung injury model, we decided to use regular FBS to avoid negatively impacting the potency of the cells or their EV. We hope this addresses this issue.

Once again, we appreciate your constructive feedback, and we are committed to addressing these concerns thoroughly in our revised manuscript. If you have any additional questions or if further clarification is needed, please do not hesitate to let us know.

Thank you for your time and consideration.

Reviewer 2 Report

Comments and Suggestions for Authors

The work by Valiukevičius et al. demonstrates therapeutic effects of placental MSC and MSC-derived EV in a mouse ASRD model on a number of readouts. The experiments are well designed and the manuscript is well written. However, the feeling I get by reading this manuscript is that the authors are over-eager to demonstrate positive effect by stressing non-significant trends. When something is not significant, it has to be concluded that there is either no effect, or that n numbers are too low. Variation in the results is pretty large and n numbers for each of the individual experiments are not mentioned, which would be necessary to evaluate the robustness of the findings. Without adapting this issue the manuscript is not acceptable for publication for me. I furthermore suggest to display individual data points in the graphs so that readers get a more fair insight in the results.

The introduction is pretty long and parts are not directly relevant for the study. For instance the part on quality controls for MSC does not link with the experimental part, and the introduction on ASRD is very extensive.

The methods are detailed and very clearly written.

Results / discussion

-          What is meant with viability in Figure 1C? Is that viability after trypsinisation?

-          Figure 1D-E: most studies show an increase in population doubling times over passages. Why would populations doubling times decrease here?

-          Figure 1F misses negative / unstained controls.

-          The authors put a lot of emphasis on the clinical grade of their MSC (for instance: No bacterial growth was observed in the cell supernatant throughout the process, and none of the tested samples were contaminated with mycoplasma. The results of the kinetic chromogenic endotoxin analysis showed that the endotoxin levels were below the acceptable limit of 5 EU/kg, which indicates that the P-MSC preparations were not contaminated with endotoxins.) This is perhaps relevant information in the methods of a clinical study, but not in the results section of an animal study.

-          The authors state in line 411 ‘P-MSC EV in beads-based flow cytometry showed the following percentages of exosomal markers: CD63 18.1 %, CD9 33.3 %, and CD81 30.7 %.’ Where can we find the data?

-          Paragraph 3.3 ‘Extracellular Vesicle Therapy Significantly Reduces Lung Injury Score’ lacks an introduction of the experiment. Clearly explain what was done.

-          Figure 3 (and others) Please indicate n number in the legend. In the text a p value of 0.06 is mentioned between P-MSC and EV therapies and in the figure 0.05 is shown. Which is correct?

-          Throughout all figures, please be consequent and show only significant p values.

-          Figure 7. If the number of Treg cells in the tissue was too small to perform reliable measurements, why show the data?

Minor comment

Page 2, line 86. The term ‘MSC transplantation’ is not really correct here. In case of IV or intratracheal injection it is more correct to speak about administration.

Author Response

Dear Reviewer,

We are grateful for your in-depth review of our manuscript. We appreciate your effort, and we are thankful for the opportunity to address the points you raised.

  1. As for the over-eagerness of our results section, we have revised the text, and excluded most of the trend interpretations.
  2. Thank you for the suggestion to include individual data points, we have adjusted our graphs accordingly.
  3. We have also shortened the ARDS and quality control section, to include only parts that are directly relevant to our study.
  4. As for the questions regarding Figure 1, we have clarified in the text that the viability is measured after trypsinization (detachment) of the cells. As for the negative controls, we are unable to add the negative controls to the graph directly, however, we have used isotype and unstained controls to set the gates for negative cells, which are shown in the figure. Finally, we would like to distinguish between the Population doubling time per passage and Cumulative Population Doublings, with the latter increasing over time, while the former is dependent on multiple factors. These factors include the source of cells for isolation, the culturing methods (culture media, seeding density, cell generations, handling technique). Most of the articles referring to the increase of the population doubling times are referring to cumulative PDT, and especially at later passages. However, there are articles demonstrating similar results (10.3389/fimmu.2020.00826), and if the PDT is not decreasing it can effectively stay at least the same (10.3390/ijms222413515)., Our approach is designed to increase the proliferation potential of the cell culture, i.e., very early passages, early trypsinization, specific cell handling techniques. It is worth mentioning, the increase of doubling time is expected in later passages.
  5. As for the comment of the text describing clinical grade MSC, we believe this is important to establish early in the research with advanced cell therapy products, since this leads to better translation in further clinical trials. Therefore, we would like to include sections describing the quality controls we have in place for our cell cultures.
  6. As for the CD markers, we have added the min-max range for the samples we have analyzed before injection to animals. We assume that a graph for this data would not add any additional information and it is better to simply describe these characteristics in text.
  7. As for paragraph 3.3, we have included a short introduction, thank you for this suggestion.
  8. We have also included the sample sizes in each legend. Thank you for noticing our error on the p-value in the text, we have corrected it.
  9. Thank you for the suggestion to show only the significant p-values, we have edited our figures.
  10. We have moved the Foxp3 gene expression graph to supplementary figures, we agree that it is not very sensitive to changes in tissue.
  11. We agree that MSC transplantation is not a fitting term, we fixed it to administration.

Again, we value your constructive feedback, and we are dedicated to addressing these concerns comprehensively in our revised manuscript. Should you have any additional questions or require further clarification, please let us know.

Thank you for dedicating your time and consideration to our work.

Round 2

Reviewer 2 Report

Comments and Suggestions for Authors

The authors improved their manuscript by providing further clarifications of the methods and results. In particular, showing individual data points and removal of the discussion of trends resulted in a more fair representation of the data to the reader. I have one final comment, and that is that in Figure 4 no individual data points are shown. The reason that the authors give is that there are too many data points (10). This is however not a valid reason. So please change this figure so it is in line with the other figures.

Author Response

Dear Reviewer,

Thank you for your final comment. We will update Figure 4 to include individual data points. 

We are grateful for your consideration.